# Effect of Municipal Solid Waste Slag on the Durability of Cementitious Composites in Terms of Resistance to Freeze–Thaw Cycling

**DOI:** 10.3390/ma16020626

**Published:** 2023-01-09

**Authors:** Marta Thomas, Agnieszka Ślosarczyk

**Affiliations:** Faculty of Civil and Transport Engineering, Institute of Building Engineering, Poznan University of Technology, 60-965 Poznań, Poland

**Keywords:** municipal solid waste incinerated slag (MSWI slag), cementitious composite, long-term behavior, durability, freeze–thaw resistance, abrasion

## Abstract

The article presents durability results for cement mortars made with the addition of slag from municipal waste incineration plants as a replacement for natural aggregate. The undertaken durability tests included frost resistance tests and evaluation of strength, microstructure, water absorption and abrasiveness before and after 150 freeze–thaw cycles. The work reveals that MSWI slag in amounts up to 50 vol. % caused deterioration in the workability and water absorption of cement mortars, regardless of the type of cement used. This, in turn, resulted in a reduction in the compressive and flexural strengths of the composites compared to mortars made with sand alone. Nevertheless, the use of a higher grade of cement, CEM I 52.5 R, resulted in an increase in compressive strength and thus increased the mortars’ frost and abrasion resistance. In addition, after the induced freeze–thaw cycles, mortars made with MSWI slag showed comparable or higher frost and abrasion resistance than those made using natural aggregate.

## 1. Introduction

It is generally known that the building industry is responsible for 40% of all energy consumption and is currently one of the most energy-intensive industries, with high carbon dioxide emissions into the atmosphere [1]. Therefore, sustainability in the construction industry has received particular attention in recent years. Directly related to this issue are 4 out of 17 Sustainable Development Goals, namely, goals 9 (industry, innovation and infrastructure), 10 (reduced inequalities), 11 (sustainable cities and communities) and 12 (responsible production and consumption) [2,3]. Following these targets involves introducing new and innovative technologies in the construction sector that will reduce energy consumption, decrease CO_2_ emissions and follow the guidelines set by the circular economy according to the development paths shown in Figure 1. The sustainable approach has also led to special attention being paid to the durability of materials, especially in the case of the cementitious binder that is most commonly used today and materials based on this binder, such as mortars and concretes. Most commonly, the durability of cementitious composites is understood today as high strength combined with high material tightness and high resistance to external factors, both chemical and physical. It is believed that the proper design of cementitious composites, taking into account the above-mentioned issues, will contribute to a longer service life of these materials and, therefore, a lower environmental impact [4]. The relevance of the issue is demonstrated by the diagram presented in Figure 2, which shows the most frequently used keywords in the last twelve years in over 2000 scientific publications. These keywords were cited at least 15 times in the publications. The size of the circle depends on the intensity of the citations, and the colors show the relationship between the individual keywords. From this, it can be seen that, most often, the durability of concrete is linked to the compressive strength of the material. This diagram also shows certain subject groups that influence the durability of the material, namely, the cement binder, cement additives such as silica fume, fly ash or chemical admixtures, the porosity of the structure and permeability. A review of the keywords in publications also shows the link between these issues and sustainability.

Ensuring the durability of cementitious composites means that the components made from them will be used longer, and the problem of recycling and reusing these materials will be postponed. Nevertheless, recent years have clearly indicated the importance of waste management, which should be in line with the current requirements of a circular economy. In this aspect, concrete and other composites based on cementitious binders show a high efficiency in recycling and reuse [5,6]. Old concrete waste, after appropriate segregation, can be reused according to EN 206 as coarse aggregate for concrete up to 30 vol. % and up to strength class C30/37. In addition, concrete is a material for which other waste materials can be used, either as a partial replacement for the cement binder (e.g., blast furnace slag, fly ash, silica fume) or as a replacement for part of the aggregate (e.g., glass cullet, brick, cork, polystyrene wastes, etc.) [7,8,9,10,11]. In the latter group, there is also the possibility of using slag from municipal waste incinerators as aggregate. Research conducted in recent years indicates that slag from waste incineration plants (MSWI) can successfully replace sand fractions in mortars and concretes up to 30 percent by volume or can be used as substructures [12,13,14,15]. However, the addition of MSWI slag has a slight effect on the deterioration of the mechanical performance and on the increased water absorbability of the cement composites due to the lower density and more porous structure of the slag compared to a natural aggregate [13,16]. In addition, a very important issue related to the disposal of MSWI slag in cement mortars and concretes is their chemical composition, especially the presence of some heavy metals, increased amounts of chlorides, sulfates or unburned carbon. In the case of increased sulfate and chloride content, we can expect the formation of salts with swelling properties, the presence of which contributes to the internal corrosion of concrete [17]. On the other hand, in the work of Ghafari et al., Ha et al. and Freeman et al. it was shown that fly ash containing high contents of unburned carbon can significantly affect a number of rheological as well as durability properties of cementitious composites, including color, the electrical conductivity of the concrete, increased water-binder ratio, compatibility with admixtures, or corrosion effects on steel rebar. It has also been noted that excessive amounts of carbon, due to its high specific surface area, can deactivate aerating admixtures, thereby causing improper aeration of concrete mixtures and leading to a lack of frost resistance of concrete. Therefore, when considering the reuse of MSWI slag in cement composites, its durability should also be taken into account. The most common applications indicated are the use of MSWI slag in fine concrete elements or as aggregate for substructures. In these solutions, the durability of the material, especially in terms of frost resistance and abrasion resistance, will be important [13,18,19,20,21].

To the best of the authors’ knowledge, information on the durability issues of cementitious composites where slag from municipal waste incinerators has been used as an aggregate is very limited. Therefore, this paper presents the results of a study on the role of MSWI slag in shaping selected durability properties of cementitious composites in physical terms, namely, resistance to cyclic freezing and thawing. It was also examined how frost action affects specific parameters of cementitious composites—precisely, water absorption, abrasion and microstructure. The study also assessed how increasing the cement grade affects the individual physico-mechanical parameters and, finally, the durability of the composites produced.

## 2. Materials and Methods

### 2.1. Cement Mortar Preparation

CEM I 42.5 R and CEM I 52.5 R Górażdże are hydraulic binders that are made by co-grinding Portland clinker (main component) and sulphate raw material (setting time regulator); both Portland cement complies with EN 197-1 [22]. Chemical properties of Górażdże products are presented in Table 1 and Table 2, while the mechanical and physical properties are collected in Table 3. Chemical analysis of the composition of CEM I 42.5 R and 52.5 R cements presented in Table 1 and Table 2 was carried out by instrumental methods according to the standard PN-EN 196-2:2013 [23].

These kinds of cement are suitable for the production of concrete mixtures containing additions of fly ash and ground blast furnace slag, allowing the optimum use of their pozzolanic and hydraulic properties. The processed slag (Figure 3a) consists of silica, metal oxides and a small amount of unburned carbon and water. Figure 3b shows the microstructure of MSWI slag. The picture clearly shows the porous structure of the slag, with numerous inclusions of unburned residues such as glass.

The EDS results with element composition are presented in Figure 4, while the chemical composition of slag is presented in Table 4. The elemental analysis confirms the presence of the oxides listed in Table 4. In addition, unburned carbon, potassium, zinc and chlorine were found in the MSWI of the slag. An important issue from the point of view of the potential use of blast furnace slag as an aggregate for mortar and concrete is the unburned carbon content, which is about 6.2 wt. % for the slag tested. Acceptable amounts of carbon in fly ash currently used as an additive for cement and concrete, according to European guidelines PN-EN 450-1 and PN-EN 197-1 [22,24], can be up to 9%. Nevertheless, large amounts of unburned carbon can adversely affect the properties of mortars and concretes. It has been reported, among other things, that large amounts of carbon have a negative effect on compatibility with chemical admixtures, mainly aeration, increase the corrosive properties of the cement matrix against steel reinforcement and increase its conductivity [18,19,20,21]. Therefore, each time the suitability of fly ash with large amounts of unburned carbon should be verified. The MSWI slag was supplied by ITPOK (Installation for Thermal Transformation of Municipal Waste in Poznań, Poland). The slag of the 0–10 mm fraction was dried using a gas burner and sieved to separate the 0–4 mm fraction. The quartz sand used for cement mortars is fine aggregate with grains up to 2 mm in diameter (KWARCMIX, Tomaszów Mazowiecki, Poland).

Sieve analyses of standard sand, slag and sand-slag mixtures used to prepare the samples are shown in Figure 5. Both slag and mixtures of fine aggregate consisting of slag and sand achieved a grain size curve very similar to that of standard sand. The sieving curves show that MSWI slag, compared to sand, contains a higher proportion of fractions above 0.125 mm, and additionally contains a 2–4 mm fraction, which accounts for about 30% of the aggregate. Mixtures of sand and MSWI slag were between the sieving curve of slag and sand.

Three types of mortars with each cement type were made. The mortars differed in proportion of aggregate. The sample naming legend is presented in Table 5. The water-to-cement ratio was constant and amounted to 0.58.

### 2.2. Determination of Consistence of Fresh Mortar

Consistency of fresh mortar was determined according to the PN-EN 1015-3 standard [25]. A standard flow table was utilized. Accordingly, the glass disc was first wiped with a wet cloth. A mold was placed centrally on the disc of the flow table (with the larger diameter down), and a filling adapter was placed on the mold. The mortar was introduced in two layers, each layer being compacted by at least 10 short strokes of the tamper. The adapter was then removed, and the excess mortar was skimmed away with a ruler, and the free area of the disc was wiped. After approximately 15 s, the mold was slowly raised vertically and removed. The flow table was subsequently jolted 15 times by rotation of the crank at constant frequency of approximately one per second. Immediately after last rotation, the diameter of the mortar was measured in two directions at right angles to one another. The mean value of diameter was then calculated.

### 2.3. Determination Flexural and Compressive Strength

Flexural and compressive strength were assessed according to EN 196-1 standard [26]. In order to ascertain the mechanical properties of the samples, molds of standard dimensions were used (40 mm wide, 40 mm high, 160 mm long). The molds were covered with anti-adhesive liquid and then filled with the cement mortars. The samples were demolded after 24 h and left for 28 and 90 days in a water environment so they could obtain adequate compression and bending strength. In order to receive reliable results, six samples of each recipe were prepared. Bending strength was determined by utilizing a hydraulic press. Each sample was weighed first. The samples were processed to break at rates of 2 mm/min, with initial value of 50 N rising to a max of 3000 N/min under stress control. Compressive strength was ascertained by means of a hydraulic press. Each sample was weighed first. The samples were processed to destruction at rates of 144 kN/min.

### 2.4. Testing of Concrete Resistance to Frost

Testing of mortar resistance to frost was carried out in accordance with PN-B-06265 standard [27]. After 28 days of maturing in water, 6 out of 12 mortar samples (with dimensions of 40 × 40 × 160 mm) were weighed and placed in a climate chamber. The samples were subjected to 150 cycles of freezing and thawing. The specimens were then reweighed and subjected to bending and compressive strength tests in accordance with point 2.3. The remaining 6 reference samples were also weighed and subjected to strength tests.

### 2.5. Determination of Water Absorption

Water absorption of mortar was determined according to the PN-85/B-04500 standard [28]. Firstly, hardened samples were dried in an oven at 105° C until constant mass is obtained. The samples were subsequently weighed with an accuracy of 1 g. The volume of the samples was then calculated. After this, the samples were placed in a container and water was gradually added. After 24 h from the moment of complete flooding of the samples, the samples were weighed. The weighing was repeated until the difference in weight within 24 h was less than 2 g. The water absorption was calculated as the percentage increase in the sample mass in relation to the sample mass before it was saturated with water.

### 2.6. Abrasion Test

The abrasion test was carried out in accordance with EN 13892-3 standard [29]. From 10 × 10 × 10 cm cubes of mortar, cubic samples with a side of 71 ± 1.5 mm were cut. After drying to constant mass at 110 ± 5 °C, the samples were weighed with an accuracy of 0.1 g and measured with an accuracy of 0.1 mm, and then they were tested using a Bohme abrader. Each specimen underwent testing for 16 cycles, each consisting of 22 revolutions. The wear resistance Bohme was determined as loss in volume, calculated from changes in mass.

### 2.7. Microstructure Research

In order to establish the mechanism of contact zone formation, research was run utilizing the following scanning microscope: TESCAN3VEGA. To observe the microstructure of the cement mortars, the samples were crumbled in order to create pieces of approximately 1 cm^2^ area with flat surfaces. All samples were taken after the strength test and dried for 24 h at a temperature of 105 °C.

## 3. Results

### 3.1. The Influence of Slag Content on Mortar Plasticity

Figure 6 shows the influence of the slag content on the consistency of the fresh mortars depending on the type of cement used. We found that mortars with higher slag content were characterized by a lower plasticity due to a higher porosity, low density and greater water demand of the slag as aggregate [13,30,31]. Similar results were also obtained by Cheng et al., who used MSWI slag with a granularity of less than 4.75 mm as a replacement for sand in amounts ranging from 10 to 40%. It was shown that mortar flows decreased from 131 mm (for mortars without MSWI) to 101 mm (for mortars with 40% MSWI at the expense of natural aggregate) [32]. Rashid and Franz, on the other hand, substituted natural aggregate completely and showed that for mortars with MSWI slag, the water requirement increased by more than 30% compared to mortars made on sand [33]. The type of cement used did not significantly affect the consistency of the mortar. The lower plasticity of mortars containing CEM 52.5 is due to the larger specific surface of cement CEM 52.5 in comparison to the specific surface of cement CEM 42.5 [34,35].

### 3.2. The Influence of Slag and Cement Grade on Density and Flexural Strength of Cement Mortars

The mortars were tested for flexural strength after 28 and 90 days. Flexural strength results and density are presented in Table 6. The density of all types of samples was comparable; nevertheless, a slight decrease in density was observed in the case of mortars, where 20 and 50 percent of the aggregate was replaced with MSWI slag. A reduction in the density of fresh mortars was also noted by Cheng and colleagues. Replacement of sand in mortars up to 40% by MSWI slag resulted in a decrease in composite density from 2248 kg/m^3^ to 1986 kg/m^3^ [32]. The study also showed that the addition of slag negatively affected the flexural strength of the composites regardless of the type of cement used. After both 28 and 90 days of curing, a decrease in the flexural strength of the mortars was observed as the MSWI slag content increased. The greatest decrease in strength was noted for mortars made with the higher-grade cement CEM 52.5. This showed that as the compressive strength increased, the cement matrix became more brittle, resulting in a decrease in flexural strength [36,37]. In addition, the introduced slag, due to its porous structure, weakened the contact zone between the aggregate and the cement binder, resulting in a decrease in mechanical performance. The deterioration of the contact zone, as pointed out by some authors, may also be due to the higher water absorption of the MSWI slag [13,16]. This is also indicated by the water absorption results shown in Figure 13. Accordingly, the absorbed water evaporates during maturation, further creating micro-cracks that can weaken the aforementioned transition zone.

### 3.3. The Influence of Slag and Cement Grade on the Compressive Strength of Cement Mortars

The compressive strengths of the tested mortars after 28 days and 90 days of curing are shown in Figure 7. All samples made with CEM I 52.5 had a higher strength resistance than did samples made of cement CEM I 42.5 after 28 days of curing. As the maturation time increased, increases in the compressive strength were observed irrespective of mortar composition. However, some differences in the behavior of mortars with MSWI slag additions were observed at longer maturation times, especially for mortars based on CEM 52.5. Compressive strength samples made of CEM 42.5 after 28 days were comparable regardless of the type of aggregate used, while samples with cement CEM 52.5 had a significant difference in strength; the samples had the highest strength on sand alone, the lowest with 50% slag content. Compressive strength after 90 days had a similar distribution—as the MSWI slag content increased, a decrease in the compressive strength of the mortars was observed, with these decreases being greater for mortars made with CEM 42.5. The decrease in compressive strength for mortars where the natural aggregate was replaced with MSWI slag was also shown in studies by other researchers. In general, a decrease in compressive strength of 2–30% was observed for every 10% replacement of sand by slag [38].

Figure 8 and Figure 9 show the results of elemental analysis of the mortars made on CEM 42.5 and CEM 52.5, respectively. A comparison of the elemental compositions of mortars on the sand and those made with the addition of MSWI slag shows that the microstructure of the cement matrix is enriched in the elements present in MSWI slag (see Figure 4). With an increasing proportion of slag, regardless of the cement used, an increase in the content of elements such as magnesium, iron, sulfur, chlorine and phosphorus was observed. In addition, the changing carbon content was noted when MSWI slag was added to the cement mortars; the presence of MSWI slag resulted in an increase in the percentage carbon content to a level of 9–11. This was several percent higher than for mortars made on pure sand. According to the literature, the increased carbon content of MSWI slag to an amount of 12% can cause a decrease in the strength parameters of mortars [17,39].

### 3.4. Frost Resistance

The results of the influence of 150 cycles of freezing and thawing on the compressive strength of the samples are shown in Figure 10. We noted that, generally, the compressive strength results after the frost test were less than the compressive strength of the reference samples; the exception was a sample with 20% slag content as aggregate made with CEM 52.5. This showed a slight increase in strength. Both samples with sand only are characterized as having the highest loss of strength after the frost test. Resistance to cyclic freeze–thaw for mortars with MSWI slag, even with the substitution of natural aggregate at 50 vol. % was very high and a slight decrease in strength was noted.

The percentage loss of strength after 150 cycles of freezing and thawing is shown in Figure 11. The highest percentage decrease was recorded among the sand-only samples based on cement CEM I 52.5 and reached 26.1%, while all samples made of cement CEM I 42.5 had a decrease of strength equal to 12%. The smallest decrease of strength was recorded among the samples made of CEM 42.5 and 50% of slag as aggregate and equaled 0.4%. The percentage strength loss of mortars containing 20% slag as aggregate and CEM 42.5 was found to be slightly higher and amounted to 1.4%, while samples made of CEM 52.5 achieved a slight (1.8%) increase in strength.

Of note, some of the samples (1 with both types of cement and 2 based on CEM 42.5) showed a slight weight gain after the freeze–thaw test, and some (2 based on CEM 52.5 and 3 with both types of cement) showed a slight weight loss. The percentage changes in the mass of the samples after the frost test are shown in Figure 12. Maximum percentage mass gain occurred in mortars with 20% of slag as aggregate based on CEM 42.5, while maximum percentage mass loss occurred in mortars based on CEM 52.5 with 50% slag as aggregate. It is worth underlining that all types of samples show a weight change of less than 5%.

The frost resistance of cementitious composites with the addition of MSWI slag has so far been a very rare issue in the literature [40,41,42]. Studies conducted by Keppert et al. have shown that there was no deterioration in the frost resistance of cement composites with MSWI slag [40]. On the contrary, when MSWI slag up to 20% was used, an improvement in the resistance to cyclic freezing–thawing of the obtained concretes was observed. The porous structure of the slag acted as an aerating admixture, increasing the space content for freezing water. Similar relationships were noted by Berg and Neal using MSWI slag as an aggregate in large concrete blocks [42] and by Jansegers in hollow construction materials [41]. In both cases, there was no decrease in strength parameters after cyclic freezing–thawing. Nevertheless, the increase in the weight of samples with MSWI slag after 150 thaw-freeze cycles is noteworthy. This may, over longer periods, contribute to the swelling of salts precipitated from the MSWI slag structure and cause internal corrosion of the concrete. This is a complex issue and should be the subject of further research. Some authors of works point out the necessity of modifying slag by precipitating chlorides, sulfates and aluminum, before using it as an aggregate for cementitious composites in order to increase the durability of final materials made with it [17,43].

### 3.5. Water Absorption

The water absorption of the reference samples and those subjected to the frost resistance test is shown in Figure 13. The water absorption of the reference samples was seen to increase with the increase in the slag content. Additionally, samples containing CEM 52.5 had higher water absorption than did samples containing CEM 42.5. The water absorption of all samples subjected to the frost resistance test was determined to be higher than the water absorption of the reference test samples. However, it is worth noting that the water absorption of the samples subjected to the frost resistance test containing CEM 52.5 and slag (samples 2 and 3) was only slightly higher (by 0.3 pp) than the water absorption of the corresponding reference samples. Moreover, samples with CEM 52.5 containing slag and subjected to frost resistance test had lower water absorption than did samples with CEM 42.5, the water absorption of which increased after the test by 0.8 and 1.6 pp. The water absorption of mortars containing only sand increased after the frost resistance test by 0.8 pp for CEM 42.5 and by 0.9 pp. for CEM 52.5.

### 3.6. Abrasion

The volume loss after the abrasion test of reference samples and samples subjected to the frost resistance test is presented in Figure 14. In general, samples with a higher slag content showed a lower abrasion resistance and a higher volume loss. Moreover, the samples containing CEM 52.5 had a higher abrasion resistance than those containing CEM 42.5. The presence of MSWI slag in cement mortars, characterized by a more porous structure, as shown in the SEM image in Figure 3b, contributes to a decrease in the compressive strength of the composites, which, in turn, has a direct impact on the abrasion performance of the cement matrix. Deterioration in the abrasion performance of cementitious composites was also observed when other recycled aggregates, such as aggregate derived from concrete rubble, were used. In this case, a much looser and more porous top layer of mortar on the recycled aggregate is responsible for the decrease in matrix durability. As a result, a weakening of the transition zone between the new cement paste and the recycled aggregate and a decrease in strength parameters is observed [44,45]. The only exception was reference samples containing only sand (1), which had the same abrasion resistance. A way to improve the abrasion performance of MSWI slag cement composites can be to improve adhesion at the aggregate-cementitious interface by using pozzolanic additives or by using randomly distributed fibers. According to the work of Wang et al., the addition of polyvinyl alcohol fibers PVA at 1.2–2.4 kg/m^3^ combined with expansive MgO can significantly improve the splitting and compressive strength of the cement composite. The addition of MgO, which reacts with water to form magnesium hydroxide, counteracts shrinkage during drying, further enhancing the effect of PVA fibers [46,47,48]. Importantly, all samples containing slag and subjected to the frost resistance test achieved higher abrasion resistance. This may be related to the increased weight of the samples after exposure to 150 freeze–thaw cycles, as shown in Figure 12. The increased content of chlorides and sulfates in the MSWI slag may lead to the formation of salt in the composites, which tightens the structure, thus leading to increased abrasion resistance. As for the samples containing only sand as aggregate, the abrasion resistance of samples utilizing CEM 42.5 decreased and the samples containing CEM 52.5 increased.

### 3.7. Microstructural Characterization of Cement Composites

Figure 15 and Figure 16 show the microstructures of the produced cement composites before and after the frost resistance test, respectively, for mortars made with CEM I 42.5 and CEM I 52.5 cement. Regardless of the cement used, all systems were characterized by a homogeneous structure, well-developed crystallites of the cement binder phases and good adhesion to natural aggregate and MSWI slag. No increased porosity of the mortars was observed in the images when sand was replaced by slag. Nevertheless, in some microphotographs, micro-cracks within the cement matrix are visible, which may be the result of the evaporation of water from the material structure. An increased amount of micro cracking was observed in the samples subjected to reduced temperatures. In addition, at higher image magnifications, deterioration of the break surfaces in the form of spalling was observed, especially for mortar with MSWI slag content. This could be the effect of salts crystallized during freeze–thaw cycles, associated with the increased presence of chlorides and sulfates in MSWI slag.

## 4. Conclusions

The main objective of the study was to determine the effect of replacing natural aggregate in cement mortars with MSWI slag up to 50% on the frost resistance of mortars and their mechanical durability as determined by mechanical properties and abrasion susceptibility. So far, this type of research has been reported quite rarely in the available literature. The study showed that cement mortars in which natural aggregate was replaced with MSWI slag at up to 50% by volume showed comparable or better resistance to cyclic freeze–thaw as mortars made entirely with natural aggregate. This is due to the porous structure of MSWI slag so that it performs a similar function to aerating admixtures, increasing the free spaces in the mortar where freezing water can increase its volume. Nevertheless, the presence of MSWI slag in mortars resulted in a decrease in density, flexural and compressive strength as well as abrasion resistance of mortars. This may be due to the lighter density and higher absorbency of the cement mortars, as well as the increased amount of carbon content resulting from the properties of the slag MSWI itself. It was shown that mortars with the addition of slag showed several percent higher carbon contents compared to mortars made on natural sand. A positive effect of MSWI slag was noted in the case of abrasiveness after cyclic freeze–thaw, recording much lower abrasiveness than mortars made on sand alone. It can be caused by the tightening of the cement matrix structure by salts precipitated during cyclic freeze–thaw. Studies have shown that increasing the grade of cement in mortars increases the durability of cement mortars made with MSWI slag.

## Figures and Tables

**Figure 1 materials-16-00626-f001:**
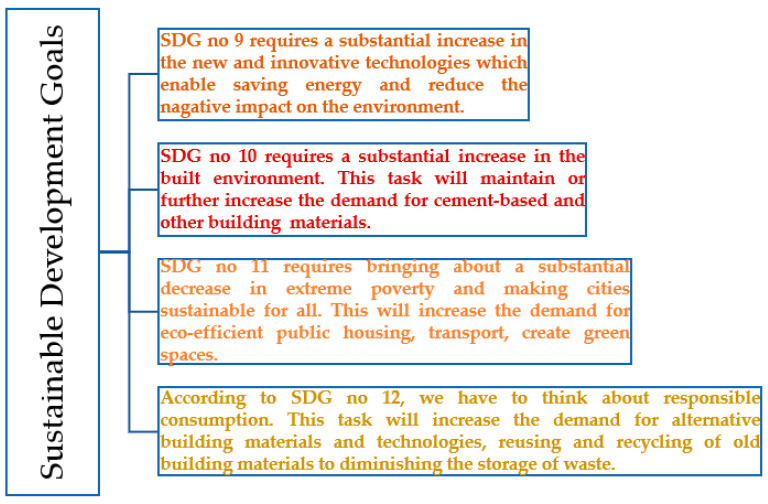
Correlation between selected sustainable development goals and development pathways in construction.

**Figure 2 materials-16-00626-f002:**
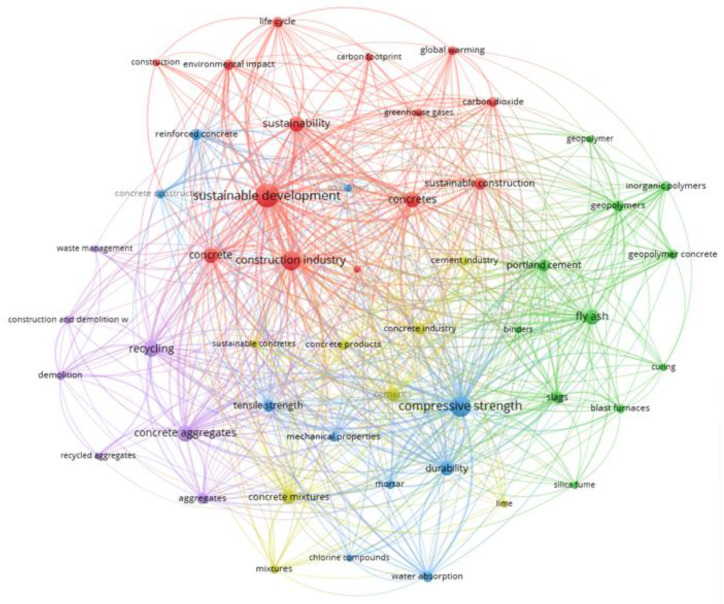
Correlation between selected sustainable development goals and development pathways in construction (based on VOSviewer, 26 November 2022).

**Figure 3 materials-16-00626-f003:**
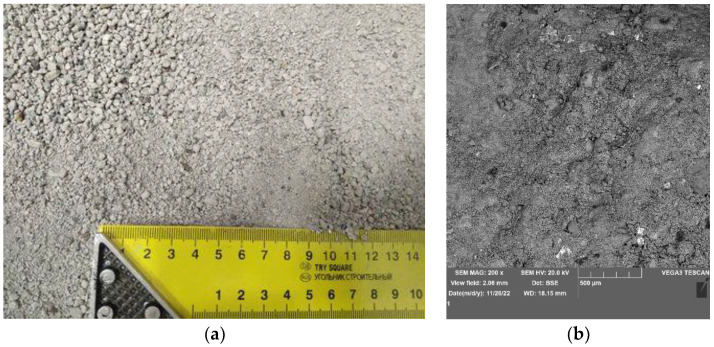
Photo (**a**) and SEM image (**b**) of the MSWI slag used for the research.

**Figure 4 materials-16-00626-f004:**
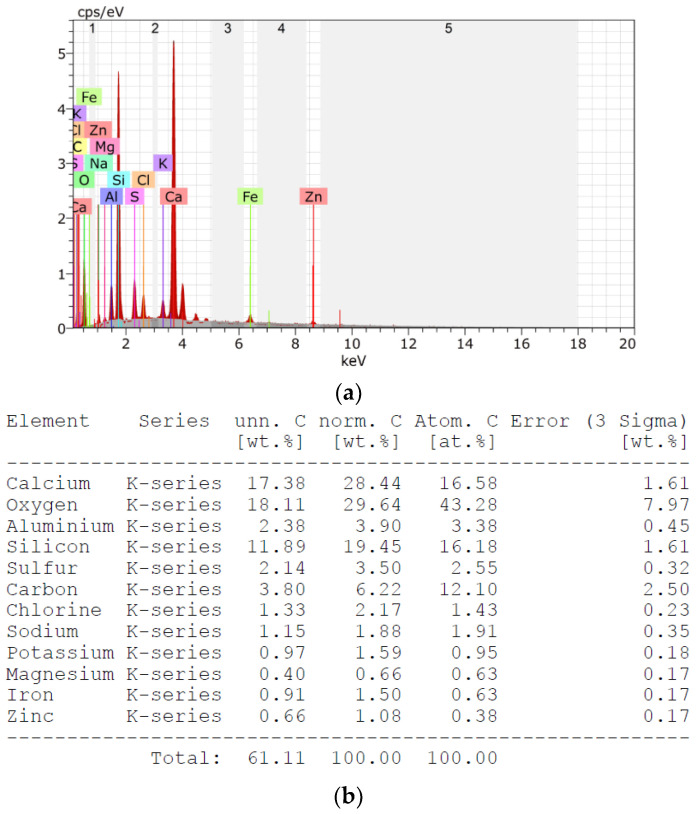
Results of EDS procedure of slag (**a**) with elemental composition (**b**).

**Figure 5 materials-16-00626-f005:**
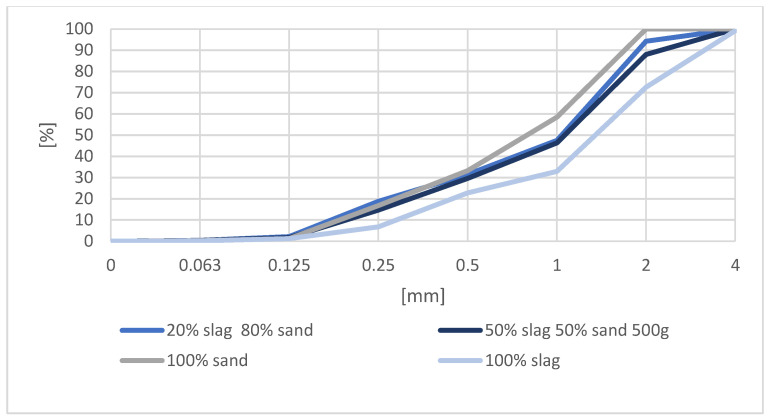
Aggregate grain size curve.

**Figure 6 materials-16-00626-f006:**
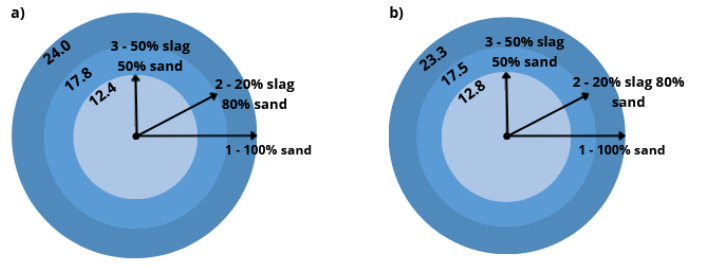
Consistency of fresh mortars based on sand and slag, with cement 42.5 (**a**) and cement 52.5 (**b**).

**Figure 7 materials-16-00626-f007:**
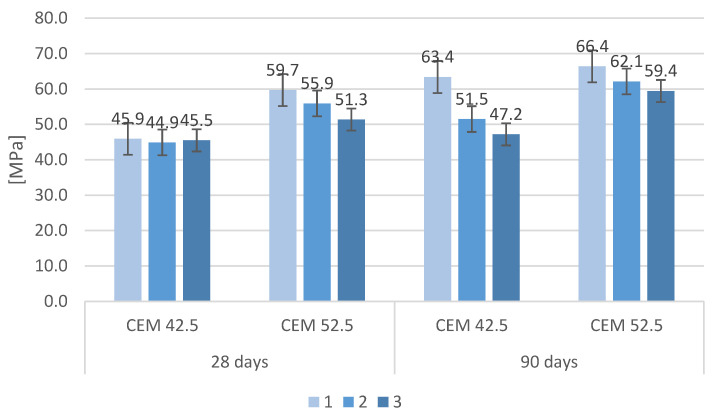
Compressive strength of cement mortars in relation to cement grade.

**Figure 8 materials-16-00626-f008:**
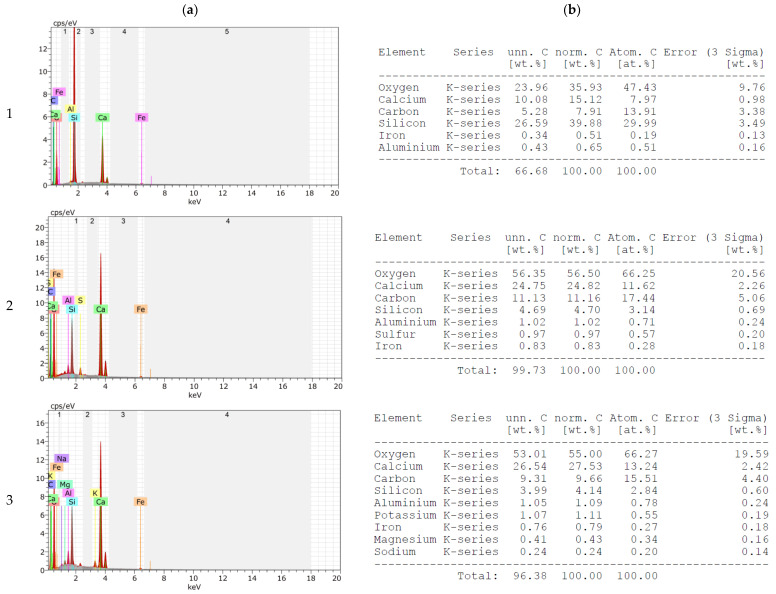
Results of EDS procedure of samples made of cement CEM I 42.5 R (**a**) with their element composition (**b**).

**Figure 9 materials-16-00626-f009:**
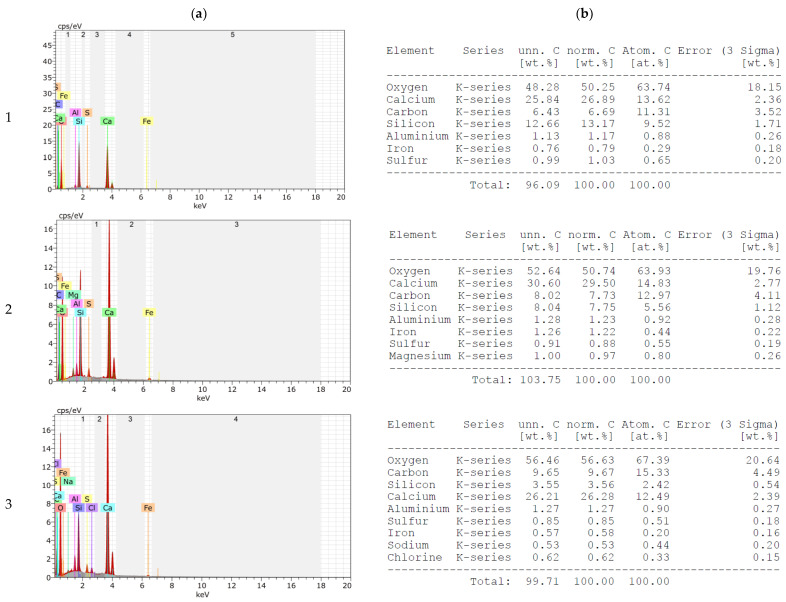
Results of EDS procedure of samples made of cement CEM I 52.5 R (**a**) with their element composition (**b**).

**Figure 10 materials-16-00626-f010:**
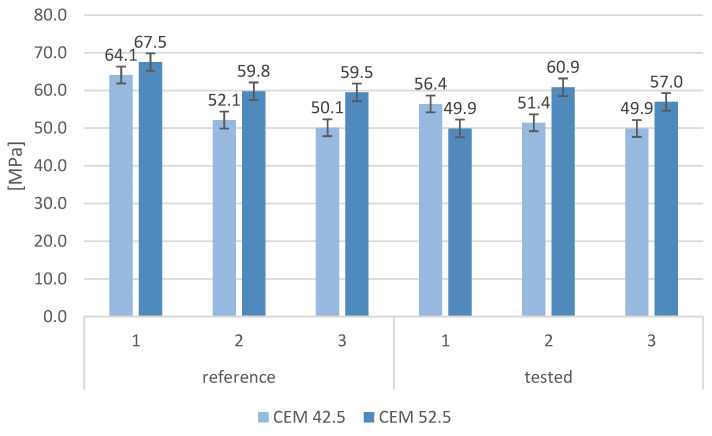
Compressive strength of samples after frost testing and compressive strength of reference samples.

**Figure 11 materials-16-00626-f011:**
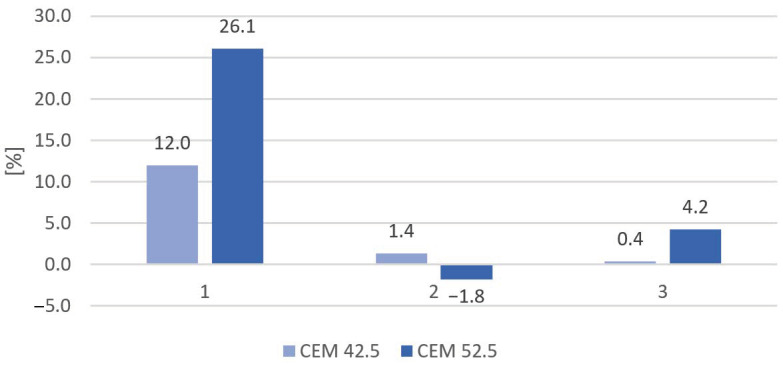
Percentage loss of strength after 150 cycles of frost resistance testing.

**Figure 12 materials-16-00626-f012:**
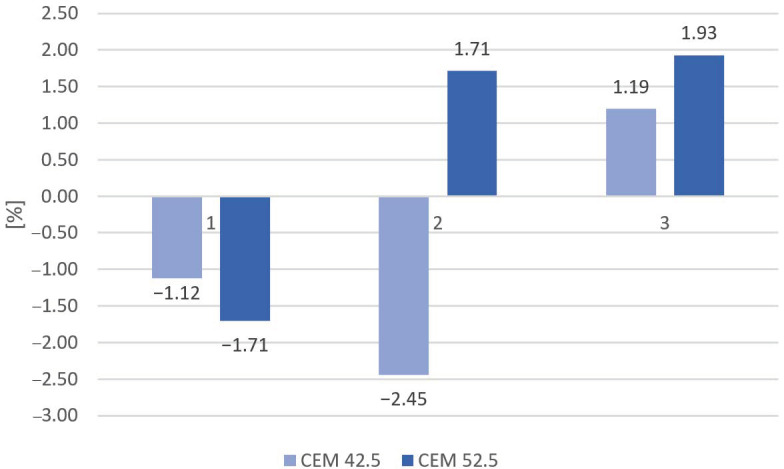
Percentage loss of mass after 150 cycles of frost resistance testing.

**Figure 13 materials-16-00626-f013:**
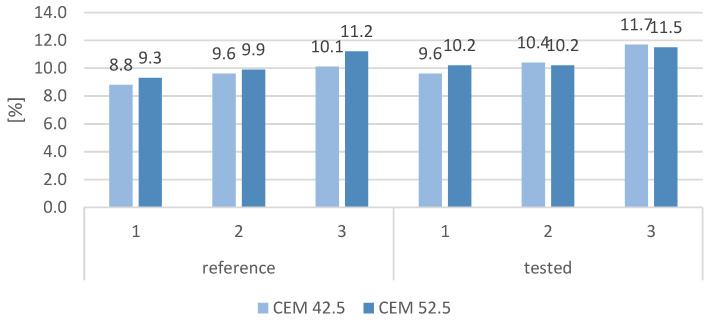
Water absorption of cement mortars, respectively, reference and subjected to cycling freezing and thawing in relation to cement grade.

**Figure 14 materials-16-00626-f014:**
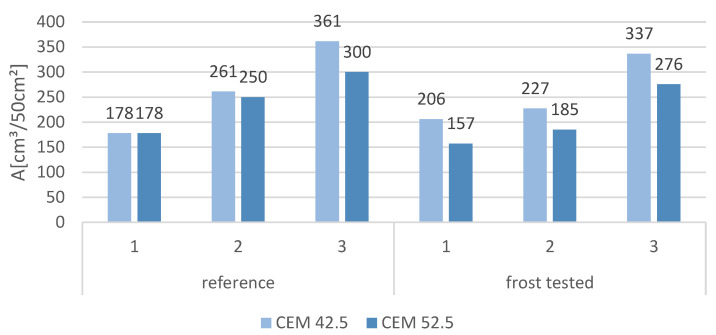
Loss of volume during abrasion testing after frost test and without frost test.

**Figure 15 materials-16-00626-f015:**
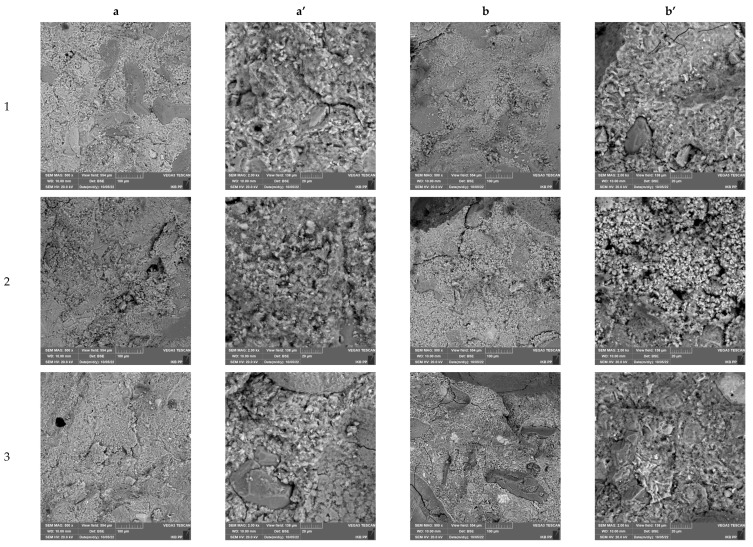
SEM images of cement mortars based on sand and slag and cement CEM 42.5 without freezing–thawing test (**a**,**a’**) and after freezing–thawing test (**b**,**b’**).

**Figure 16 materials-16-00626-f016:**
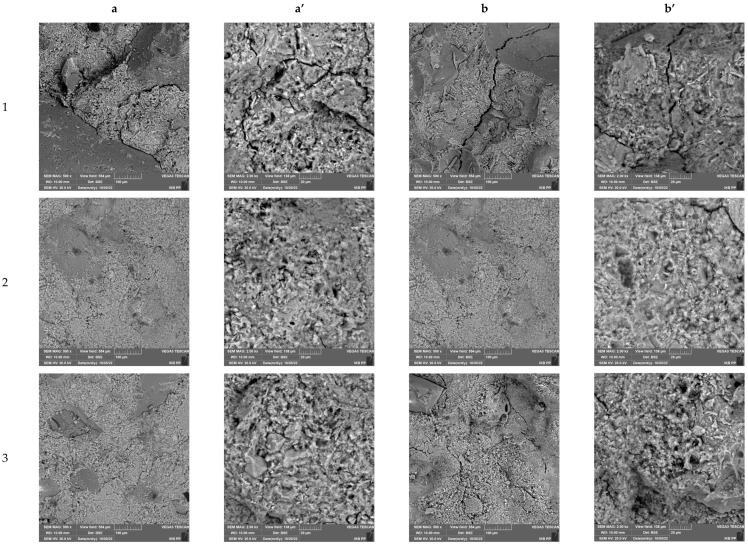
SEM images of cement mortars based on sand and slag and cement CEM 52.5 without freezing–thawing test (**a**,**a’**) and after freezing–thawing test (**b**,**b’**).

**Table 1 materials-16-00626-t001:** Chemical properties of cement.

Type of Sample	Chemical Properties				
	SO_3_ [%]	Cl^−^ [%]	Roasting Loss [%]	Insoluble Residues [%]	NaO_eq_ [%]
CEM I 42.5 R	2.89	0.061	2.54	1.93	0.61
CEM I 52.5 R	2.74	0.052	2.73	0.50	0.57

**Table 2 materials-16-00626-t002:** Chemical composition of cement.

Type of Sample	Chemical Components [%]							
	SiO_2_	Al_2_O_3_	Fe_2_O_3_	CaO	MgO	SO_3_	NaO_2_	K_2_O
CEM I 42.5 R	17.6	4.3	3.9	67.2	1.5	3.8	0.2	0.9
CEM I 52.5 R	20.2	5.5	2.2	64.8	1.7	2.5	0.2	0.7

**Table 3 materials-16-00626-t003:** Mechanical and physical properties of cement.

Type of Sample	Mechanical Properties		Physical Properties			
	Compressive Strength after 2 Days [MPa]	Compressive Strength after 28 Days [MPa]	Setting Time [min]	Water to Normal Consistency [%]	Volume Constancy [mm]	Specific Surface [cm^2^/g]
CEM I 42.5 R	27.2	59.3	203	27.4	0.6	3957
CEM I 52.5 R	34.4	68.2	184	28.8	0.3	4830

**Table 4 materials-16-00626-t004:** Chemical composition of MSWI slag.

Chemical Composition of Slag [%]							
SiO_2_	CaO	Fe_2_O_3_	Al_2_O_3_	Na_2_O	MgO	P_2_O_5_	SO_3_
44	21	14	10	4	3	2	2

**Table 5 materials-16-00626-t005:** Legend of mortar’s type.

Type of Sample		Aggregate	
		Sand [%]	Slag [%]
CEM 42.5	1	100	0
	2	80	20
	3	50	50
CEM 52.5	1	100	0
	2	80	20
	3	50	50

**Table 6 materials-16-00626-t006:** Flexural strength and density of cement mortars after 28 and 90 days of curing.

Type of Sample		Flexural Strength (FS) and Density (D) after							
		28 Days				90 Days			
		FS [MPa]	Standard Deviation	D [kg/m^3^]	Standard Deviation	FS [MPa]	Standard Deviation	D [kg/m^3^]	Standard Deviation
CEM 42.5	1	7.1	0.538	2260	16	8.6	0.617	2282	10
	2	6.7	0.509	2252	12	6.8	0.624	2240	14
	3	7.4	0.186	2236	16	7.1	0.501	2221	16
CEM 52.5	1	8.4	0.386	2286	14	5.9	0.404	2294	11
	2	7.5	0.371	2282	21	6.1	0.497	2240	18
	3	6.9	0.290	2232	12	5.9	0.526	2258	20

## Data Availability

The data presented in this study are available on request from the corresponding author.

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
