# Peer review of "Effect of Municipal Solid Waste Slag on the Durability of Cementitious Composites in Terms of Resistance to Freeze–Thaw Cycling"

_materials, 2023, doi:10.3390/ma16020626_

Round 1

Reviewer 1 Report

Ms. Ref. No.:  materials-2068226-peer-review-v1

Effect of municipal solid waste slag on the durability of cementitious composites in terms of resistance to freeze-thaw cycling

Reviewer comments:

SUMMARY

The manuscript deals with a investigation on the utilization of municipal solid waste slag and its resistance to freeze-thaw cycling. This is a topic that has not been widely covered in the literature, therefore, this subject is of great interest, but it is somehow limited in the analysis and application of these results.

MAIN IMPRESSIONS

This paper has an undeniable practical usefulness. However, from a scientific point of view, the following issues must be addressed: i) Discussion should be included; ii) References regarding resistance to freeze-thaw cycling of cement-based materials with municipal solid waste slag should be added and discussed in deep; iii) Chemical composition of the municipal solid waste slag should also be discussed taking into account the findings given in the literature.

This manuscript does follow completely the mdpi format. You should adapt this manuscript to the mdpi rules. For instance, check the references’ style and the Figures’ citation in the text (Lines 33 and 42).

MORE DETAILED COMMENTS

Line 6 - Abstract: CEM I 52,5, R or N? In line 88, CEM I 42,5 R and CEM I 52,5 R are mentioned.

Lines 26, 30 and so on: This manuscript does follow completely the mdpi format, could you please check the references’ style?

Line 88: CEM I 42.5 R and CEM I 52.5 R Górażdże are Portland cements. Could you please add the standard. Probably, this should be EN 197-1:2011 (Mater. Const. 62(307), 425–430. https://doi.org/10.3989/mc.2012.07711).

Line 88: Chemical composition of CEM I 42.5 R and CEM I 52.5 R Górażdże Portland cements should be added. In particular, a high amount of unburned carbon is responsible for the loss on ignition represents an undesirable constituent of some ashes to be utilized in the reinforced concrete construction. Therefore, it should be mentioned in the introduction. The problem is that the unburned carbon in slags and ashes has several detrimental effects on the concrete. Especially, it increases the electrical conductivity of the concrete, changes the color of mortar and concrete (they may appear black), etc. Moreover, the water/(cement+addition) ratio, needed to obtain a cement paste with a required rheological properties or consistency, is higher forsome slag or ashes with a high carbon content, increasing the corrosivity of metallic parts incorporated in the concrete. Finally, it causes a poor air entrainment behavior and mixture segregation. The following papers deal with this topic:

·       Freeman, E., Gao, Y-M., Hurt, R. and Suuberg, E.: 1997, Interactions of carbon-containing fly ash with commercial air-entraining admixtures for concrete, Fuel, 76, no. 8, 761–765. https://doi.org/10.1016/S0016-2361(96)00193-7

·       Ha, T.H., Muralidharan, S., Bae, J.H., Ha, Y.C., Lee, H.G., Park, K.W. and Kim, D.K.: 2005, Effect of unburnt carbon on the corrosion performance of fly ash cement mortar, Construction and Building Materials, 19, 509–515. https://doi.org/10.1016/j.conbuildmat.2005.01.005

·       Ehsan Ghafari, Seyedali Ghahari, Dimitri Feys, Kamal Khayat, Aasiyah Baig, Raissa Ferron. Admixture compatibility with natural supplementary cementitious materials, Cement and Concrete Composites, Volume 112, 2020, 103683, https://doi.org/10.1016/j.cemconcomp.2020.103683

·       S. Lim, W. Lee, H. Choo, C. Lee, Utilization of high carbon fly ash and copper slag in electrically conductive controlled low strength material, Construction and Building Materials, Volume 157, 2017, Pages 42-50, https://doi.org/10.1016/j.conbuildmat.2017.09.071

Lines 89-92: This is obvious. It can be deleted.

Line 93: cement strength class (compression strength after 28 days equal to 42.5 MPa and 52.5, respectively)? Are you sure? Could you please check it carefully?

Line 93: Could you please add a table with the mechanical and physical characteristics of CEM I 42.5 R and CEM I 52.5 R Górażdże Portland cements?

Line 95: Could you please add a table with the chemical composition of slag?

Line 95: “It consists of a small amount of unburned combustibles”? What kind of combustibles”?

Line 98: Could you please add the granulometric slag curve? (sieved to separate the 0-4 mm fraction).

Line 99: Could you please add the granulometric sand curve? (sand used for cement mortars is fine aggregate with grains up to 2 mm in diameter).

Line 99: Could you please add the nature of the sand (siliceous/calcareous/ etc.)?

Line 102: Why have you decided to use a water to cement ratio of 0.58? This is different to the water to cement ratio of 0.50 specified in EN 196-1.

Line 120: Have you follow the EN 196-1 standard to made the 40 mm wide, 40 mm high, 160 mm long specimens?

Line 126: destruction?

Line 127: Could you please add a table with the chemical composition of the debris?

(…the debris was retained separately for the subsequent chemical testing.)

Line 130: Could you please add a table with the chemical composition of the debris? (…debris was retained separately for the subsequent chemical testing.)

Line 131: 2.4. Testing of concrete resistance to frost è Could you please add a table with the concrete mix design?

Line 133: mortar samples or concrete?

Line 160: ”3. Results”. Discussion is missing. You should compare and discuss your work with the one performed by other authors. Discussion should be deep. Some references should be mentioned in the discussion, and they should be discussed in deep.

Please, check similar results in the literature and discuss them.

Line 285: Conclusions should highlight the novelty of the work. Please, rewrite the conclusions.

Line 320: References must follow the magazine format. Please, check it.

References

In the text, reference numbers should be placed in square brackets [ ] and placed before the punctuation; for example [1], [1–3] or [1,3]. For embedded citations in the text with pagination, use both parentheses and brackets to indicate the reference number and page numbers; for example [5] (p. 10), or [6] (pp. 101–105).

References must be numbered in order of appearance in the text (including citations in tables and legends) and listed individually at the end of the manuscript. Include the digital object identifier (DOI) for all references where available.

1.        Author 1, A.B.; Author 2, C.D. Title of the article. Abbreviated Journal Name Year, Volume, page range.

RECOMMENDATION

In conclusion, Major changes have been proposed.

Author Response

Dear Reviewer, Thank You for Your insightful review of our work, which contributed to a better understanding of the scientific problems related to the subject of the publication and will help with the elimination of potential errors in the future.We would also like to express our gratitude for the revision of our manuscript and the opportunity to re-submit it, incorporating all of the Referees’ suggestions. Our comments and changes are noted below, and are marked in yellow in the manuscript.

SUMMARY

The manuscript deals with a investigation on the utilization of municipal solid waste slag and its resistance to freeze-thaw cycling. This is a topic that has not been widely covered in the literature, therefore, this subject is of great interest, but it is somehow limited in the analysis and application of these results.

MAIN IMPRESSIONS

This paper has an undeniable practical usefulness. However, from a scientific point of view, the following issues must be addressed: i) Discussion should be included; ii) References regarding resistance to freeze-thaw cycling of cement-based materials with municipal solid waste slag should be added and discussed in deep; iii) Chemical composition of the municipal solid waste slag should also be discussed taking into account the findings given in the literature.

This manuscript does follow completely the mdpi format. You should adapt this manuscript to the mdpi rules. For instance, check the references’ style and the Figures’ citation in the text (Lines 33 and 42).

MORE DETAILED COMMENTS

Query 1: Line 6 - Abstract: CEM I 52,5, R or N? In line 88, CEM I 42,5 R and CEM I 52,5 R are mentioned.

Answer 1: Thank you very much for comment, we have added this information in article.

Query 2:  Lines 26, 30 and so on: This manuscript does follow completely the mdpi format, could you please check the references’ style?

Answer 2: Thank you for comment, we have changed the reference style.

Query 3: Line 88: CEM I 42.5 R and CEM I 52.5 R Górażdże are Portland cements. Could you please add the standard. Probably, this should be EN 197-1:2011 (Mater. Const. 62(307), 425–430. https://doi.org/10.3989/mc.2012.07711).

Answer 3: Thank you for comment, we added the information about the standard.

Query 4: Line 88: Chemical composition of CEM I 42.5 R and CEM I 52.5 R Górażdże Portland cements should be added. In particular, a high amount of unburned carbon is responsible for the loss on ignition represents an undesirable constituent of some ashes to be utilized in the reinforced concrete construction. Therefore, it should be mentioned in the introduction. The problem is that the unburned carbon in slags and ashes has several detrimental effects on the concrete. Especially, it increases the electrical conductivity of the concrete, changes the color of mortar and concrete (they may appear black), etc. Moreover, the water/(cement+addition) ratio, needed to obtain a cement paste with a required rheological properties or consistency, is higher forsome slag or ashes with a high carbon content, increasing the corrosivity of metallic parts incorporated in the concrete. Finally, it causes a poor air entrainment behavior and mixture segregation. The following papers deal with this topic:

  • Freeman, E., Gao, Y-M., Hurt, R. and Suuberg, E.: 1997, Interactions of carbon-containing fly ash with commercial air-entraining admixtures for concrete, Fuel, 76, no. 8, 761–765. https://doi.org/10.1016/S0016-2361(96)00193-7
  • Ha, T.H., Muralidharan, S., Bae, J.H., Ha, Y.C., Lee, H.G., Park, K.W. and Kim, D.K.: 2005, Effect of unburnt carbon on the corrosion performance of fly ash cement mortar, Construction and Building Materials, 19, 509–515. https://doi.org/10.1016/j.conbuildmat.2005.01.005
  • Ehsan Ghafari, Seyedali Ghahari, Dimitri Feys, Kamal Khayat, Aasiyah Baig, Raissa Ferron. Admixture compatibility with natural supplementary cementitious materials, Cement and Concrete Composites, Volume 112, 2020, 103683, https://doi.org/10.1016/j.cemconcomp.2020.103683
  • S. Lim, W. Lee, H. Choo, C. Lee, Utilization of high carbon fly ash and copper slag in electrically conductive controlled low strength material, Construction and Building Materials, Volume 157, 2017, Pages 42-50, https://doi.org/10.1016/j.conbuildmat.2017.09.071

Lines 89-92: This is obvious. It can be deleted.

Answer 4: Thank you, we have added chemical composition of cement. As for MSWI slag, it does indeed contain a fair amount of unburned carbon. This data is also cited by other researchers. Nevertheless, in this case, where we are using it as an aggregate, in amounts up to 50% by volume, we have not noticed any effect of this slag and unburned carbon on the mechanical properties and durability of the composite. The increased absorbability is rather due to the porous nature of the slag, we have included a photo of the slag surface structure in the article. Nevertheless, as suggested by the reviewer, we supplemented the introduction with information on the MSWI slag and the content of unburned carbon, which can affect other properties of cement composites.

Query 5: Line 93: cement strength class (compression strength after 28 days equal to 42.5 MPa and 52.5, respectively)? Are you sure? Could you please check it carefully?

Answer 5: Thank you very much for comment. Indeed, the composition of mortars has been changed from the requirements of EN 196-1, a higher water content has been used due to the higher water demand of MSWI aggregates. Therefore, we cannot directly refer to the standard grades of cement. Specific strength data have been included in the text.

Query 6: Line 93: Could you please add a table with the mechanical and physical characteristics of CEM I 42.5 R and CEM I 52.5 R Górażdże Portland cements?

Answer 6: Thank you for comment, we have added mechanical and physical characteristics of cement.

Query 7: Line 95: Could you please add a table with the chemical composition of slag?

Answer 7: Thank you for comment, we have added chemical composition of slag.

Query 8: Line 95: “It consists of a small amount of unburned combustibles”? What kind of combustibles”?

Answer 8: Thank you for comment. We meant unburned carbon. Changes have been made in the text.

Query 9: Line 98: Could you please add the granulometric slag curve? (sieved to separate the 0-4 mm fraction).

Answer 9: Thank you for comment, we have added granulometric sand curve.

Query 10: Line 99: Could you please add the granulometric sand curve? (sand used for cement mortars is fine aggregate with grains up to 2 mm in diameter).

Answer 10: Thank you for comment, we have added granulometric sand curve.

Query 11: Line 99: Could you please add the nature of the sand (siliceous/calcareous/ etc.)?

Answer 11: Thank you for comment, we defined the nature of sand in the text we used quartz sand.

Query 12: Line 102: Why have you decided to use a water to cement ratio of 0.58? This is different to the water to cement ratio of 0.50 specified in EN 196-1.

Answer 12: Thank you for comment . Due to the higher water requirement of the slag, after a series of preliminary studies, we decided to increase the water to cement ratio. Even with an increased amount of water, sand-only samples meet the strength requirements.

Query 13: Line 120: Have you follow the EN 196-1 standard to made the 40 mm wide, 40 mm high, 160 mm long specimens?

Answer 13: Thank you for comment. Yes, tests were conducted in accordance with EN-196-1 standard.

Query 14: Line 126: destruction?

Answer 14: Thank you for comment, we have changed the word to break.

Query 15: Line 127: Could you please add a table with the chemical composition of the debris?

(…the debris was retained separately for the subsequent chemical testing.)

Query 16: Line 130: Could you please add a table with the chemical composition of the debris? (…debris was retained separately for the subsequent chemical testing.)

Answer 15 and 16: Mortar fragments after destructive testing were left for microstructure analysis, and we additionally performed elemental analysis performed using an EDS probe, which is additionally included in the paper.

Query 17: Line 131: 2.4. Testing of concrete resistance to frost è Could you please add a table with the concrete mix design?

Query 18: Line 133: mortar samples or concrete?

Answer 17 and 18: Thank you for comment. We only tested mortars and changes have been made in the text

Query 19: Line 160: ”3. Results”. Discussion is missing. You should compare and discuss your work with the one performed by other authors. Discussion should be deep. Some references should be mentioned in the discussion, and they should be discussed in deep.

Please, check similar results in the literature and discuss them.

Answer 19: Thank you for comment. Research on this topic is limited, but we have nevertheless added a discussion with other works.

Query 20: Line 285: Conclusions should highlight the novelty of the work. Please, rewrite the conclusions.

Answer 20: Thank you for comment. We have rearranged the conclusions as suggested.

Query 21: Line 320: References must follow the magazine format. Please, check it.

Answer 21: Thank you for comment, we have changed the reference style.

Yours sincerely,

Prof of PUT Agnieszka Ślosarczyk

Reviewer 2 Report

The content has merit, but the following remarks should be carefully answered in the adapted version:

1) The abstract should be organized.

2) English typos and grammatical should be revised.

3) Please double check all the equations. Some parameters need to be defined.

4) Authors are encouraged to discuss other kinds of works on concrete structures based on the following recent works: [(a) “Stability and dynamic analyses of SW-CNT reinforced concrete beam resting on elastic-foundation”; (b) “Machine learning models for predicting the compressive strength of concrete containing nano silica”, Computers and Concrete, 30(1), 33-42. DOI: https://doi.org/10.12989/cac.2022.30.1.033].

5) Figs. 7 and 8 should be more discussed.

6) The conclusion should be revised, including the advantage of the presented technique.

Author Response

Dear Reviewer, Thank You for Your insightful review of our work, which contributed to a better understanding of the scientific problems related to the subject of the publication and will help with the elimination of potential errors in the future.We would also like to express our gratitude for the revision of our manuscript and the opportunity to re-submit it, incorporating all of the Referees’ suggestions. Our comments and changes are noted below, and are marked in yellow in the manuscript.

Query 1: The abstract should be organized.

Answer 1: Thank you for comment. The abstract was re-written.

Query 2: English typos and grammatical should be revised.

Answer 2: Thank you for comment. The article was checked for linguistic correction.

Query 3: Please double check all the equations. Some parameters need to be defined.

Answer 3: Thank you for your comment. All studies have been conducted according to current standards. The numbers of the standards have been completed in the text.

Query 4: Authors are encouraged to discuss other kinds of works on concrete structures based on the following recent works: [(a) “Stability and dynamic analyses of SW-CNT reinforced concrete beam resting on elastic-foundation”; (b) “Machine learning models for predicting the compressive strength of concrete containing nano silica”, Computers and Concrete, 30(1), 33-42. DOI: https://doi.org/10.12989/cac.2022.30.1.033].

Answer 4: Thank you for your comment. We added a discussion of the results with other published data.

Query 5:  Figs. 7 and 8 should be more discussed.

Answer 5: Thank you for your comment. We added a discussion of the results with other published data.

Query 6: The conclusion should be revised, including the advantage of the presented technique.

Answer 6: Thank you for your attention, the conclusions have been rewritten.

Yours sincerely,

Prof of PUT Agnieszka Ślosarczyk

Reviewer 3 Report

In the paper, authors investigate the effect of the addition of steel slag as replacement for natural aggregate.The current work has lack of enough literature support and the novelty of the investigation is the significant concern. In my opinion this paper needs major revision.

Comment (1): Many researchers studied the effect of steel slag on the mechanical properties and durability of mortar. Hence, the novelty of the investigation should be explained.

Comment (2): Please compare the results with the existing.

Comment (3): The results need more deep explanation. Please give physical explanation on the effect of steel slag on the durability and the mechanical strength of the mortar.

Comment (4): Is it possible to generalize the conclusions to other steel slag types or they are particular to the type you considered.

Author Response

Dear Reviewer, Thank You for Your insightful review of our work, which contributed to a better understanding of the scientific problems related to the subject of the publication and will help with the elimination of potential errors in the future.We would also like to express our gratitude for the revision of our manuscript and the opportunity to re-submit it, incorporating all of the Referees’ suggestions. Our comments and changes are noted below, and are marked in yellow in the manuscript.

In the paper, authors investigate the effect of the addition of steel slag as replacement for natural aggregate.The current work has lack of enough literature support and the novelty of the investigation is the significant concern. In my opinion this paper needs major revision.

Query 1: Many researchers studied the effect of steel slag on the mechanical properties and durability of mortar. Hence, the novelty of the investigation should be explained.

Answer 1: Thank you for your attention, the article uses slag from the combustion of municipal waste. The physico-chemical characteristics of the slag were added to the article. In addition, the purpose of the study was highlighted.

Query 2: Please compare the results with the existing.

Answer 2: Thank you for your comment, we have added a discussion of the results based on publications in the subject area.

Query 3: The results need more deep explanation. Please give physical explanation on the effect of steel slag on the durability and the mechanical strength of the mortar.

Answer 3: Thank you for your comment, we have added a discussion of the results based on unit publications in the subject area.

Query 3: Is it possible to generalize the conclusions to other steel slag types or they are particular to the type you considered.

Answer 4: Thank you for your comment. Indeed, the topic of using slag from various industries as aggregate for mortar and concrete has been a particularly important issue recently due to the circular economy. Nevertheless, slag from municipal waste incineration plants differs from steel furnace slag in its potential composition, which really depends on the type of garbage burned, its energy content, chemical composition, and the time of year. This affects the properties of the slag, so its composition can vary from one incinerator to another, and each time, in our opinion, control tests of mortars and concrete mixtures using it should be carried out.

Yours sincerely,

Prof of PUT Agnieszka Ślosarczyk

Reviewer 4 Report

This work focused on the sustainable aspect of the construction material and made a significant try. Before its acceptance, it is recommended to consider the following comments and make corresponding revisions.

1. At the end of the abstract, the authors should highlight the significance and contribution of this work.

2. Fig. 1 presented the correlation between selected sustainable development goals. However, I cannot get the relationship among these goals from the figure.

3. Regarding to Fig. 2, the number of items should be reduced, and the main characteristics should be highlighted.

4. The figures are wrongly numbered in sequence, please check.

5. In this work, the size distribution of slag and sand is different from each other, which may contribute to the difference in the compressive strength, flexural strength, etc. The authors should cover this factor in the discussion part.

6. The figure 3 and 4 (compressive strength) should be incorporated with the deviation value.

7. In section 3.4, the authors should give some explanations about mechanism behind the strength and mass loss. Just presenting the experimental results is not enough. It well known that the F-T resistance of concrete is significantly affected the air void system of concrete, however, no air-void information can be found in this study. At least the related studies should be added to enrich the statement of mechanisms and explanations of the slag effects. The studies Pore structural and fractal analysis of the effects of MgO reactivity and dosage on permeability and F–T resistance of concrete; Comparison between the influence of finely ground phosphorous slag and fly ash on frost resistance, pore structures and fractal features of hydraulic concrete can well improve this part.

8. For the abrasion resistance, the discussion on the effects of slag is poor, the related study Influences of MgO and PVA fiber on the abrasion and cracking resistance, pore structure and fractal features of hydraulic concrete can improve.

9. What is relationship between the results in section 3.7 and those in previous sections?

Author Response

Dear Reviewer, Thank You for Your insightful review of our work, which contributed to a better understanding of the scientific problems related to the subject of the publication and will help with the elimination of potential errors in the future.We would also like to express our gratitude for the revision of our manuscript and the opportunity to re-submit it, incorporating all of the Referees’ suggestions. Our comments and changes are noted below, and are marked in yellow in the manuscript.

This work focused on the sustainable aspect of the construction material and made a significant try. Before its acceptance, it is recommended to consider the following comments and make corresponding revisions.

Query 1:  At the end of the abstract, the authors should highlight the significance and contribution of this work.

Answer 1: Thank you for your comment. The abstract has been edited.

Query 2:  Fig. 1 presented the correlation between selected sustainable development goals. However, I cannot get the relationship among these goals from the figure.

Answer 2: Thank you for your comment. The sustainability goals that the authors believe most influence the development of sustainable construction and the need to move toward a circular economy have been selected. The goals are listed in the text; due to the size of the graphic, their names are no longer placed directly on the graphic.

Query 3:  Regarding to Fig. 2, the number of items should be reduced, and the main characteristics should be highlighted.

Answer 3: Thank you very much for comment, we have corrected the figure.

Query 4:  The figures are wrongly numbered in sequence, please check.

Answer 4: Thank you very much for comment, we have corrected the numbering.

Query 5:  In this work, the size distribution of slag and sand is different from each other, which may contribute to the difference in the compressive strength, flexural strength, etc. The authors should cover this factor in the discussion part.

Answer 5: Thank you very much for comment. The combination of sand and slag MSWI has a favorable effect on the grain size distribution of the mortar, as can be seen in the added sieving curves for sand and slag, and the combined systems. Sand tightens the structure of the slag, the content of fractions above 2 mm is about 30%. The sieving curves indicate that the grain size of the aggregate mixture differs slightly from that of the sand. The increased abrasiveness for mortars with MSWI slag is due to the porous structure of the slag rather than its granulation.

Query 6:  The figure 3 and 4 (compressive strength) should be incorporated with the deviation value.

Answer 6: Thank you very much for comment, we added deviation value.

Query 7:  In section 3.4, the authors should give some explanations about mechanism behind the strength and mass loss. Just presenting the experimental results is not enough. It well known that the F-T resistance of concrete is significantly affected the air void system of concrete, however, no air-void information can be found in this study. At least the related studies should be added to enrich the statement of mechanisms and explanations of the slag effects. The studies Pore structural and fractal analysis of the effects of MgO reactivity and dosage on permeability and F–T resistance of concrete; Comparison between the influence of finely ground phosphorous slag and fly ash on frost resistance, pore structures and fractal features of hydraulic concrete can well improve this part.

Answer 7: Thank you very much for comment. We have added a discussion of the topic in this part of the article.

Query 8:  For the abrasion resistance, the discussion on the effects of slag is poor, the related study Influences of MgO and PVA fiber on the abrasion and cracking resistance, pore structure and fractal features of hydraulic concrete can improve.

Answer 8: Thank you very much for comment. We have added a discussion of the topic in this part of the article.

Query 9:  What is relationship between the results in section 3.7 and those in previous sections?

Answer 9: SEM images show a homogeneous structure of the cement composite regardless of the cement used and the replacement of part of the natural aggregate by MSWI slag. At the same time, the microstructure of the composite after cyclic freeze-thaw treatment is shown. The MSWI slag photos show swelling of the cement matrix, which may be the result of salts crystallized during freeze-thaw cycles associated with the increased presence of chlorides and sulfates in the MSWI slag.

Yours sincerely,

Prof of PUT Agnieszka Ślosarczyk

Round 2

Reviewer 1 Report

Lines 98-100: Could you please add the testing methods?

Line 104: Could you please add the reference (EN 197-1)? EN 197-1:2011. Cement - Part 1: Composition, specifications and conformity criteria for common cement. European Committee for Standardization (CEN): Brussels, Belgium, 2011.

Line 104:

Data in Table 3. “Chemical composition of cement” should not be a range of values.

Calcium sulfate in Table 3 should be expressed as SO3.

Chemical composition of cement should be expressed as percentage of CaO, SiO2, Fe2O3, Al2O3, and so on.

Could you please add the testing methods?

Line 126: thesamples.

Lines 107-110: This is obvious. It can be deleted.

Line 124: C content in Fig 4b is about 6%. In particular, a high amount of unburned carbon is responsible for the loss on ignition represents an undesirable constituent of some ashes to be utilized in the reinforced concrete construction. Therefore, it should be mentioned in the introduction. The problem is that the unburned carbon in slags and ashes has several detrimental effects on the concrete. Especially, it increases the electrical conductivity of the concrete, changes the color of mortar and concrete (they may appear black), etc. Moreover, the water/(cement+addition) ratio, needed to obtain a cement paste with a required rheological properties or consistency, is higher forsome slag or ashes with a high carbon content, increasing the corrosivity of metallic parts incorporated in the concrete. Finally, it causes a poor air entrainment behavior and mixture segregation. The following papers deal with this topic:

·       Freeman, E., Gao, Y-M., Hurt, R. and Suuberg, E.: 1997, Interactions of carbon-containing fly ash with commercial air-entraining admixtures for concrete, Fuel, 76, no. 8, 761–765. https://doi.org/10.1016/S0016-2361(96)00193-7

·       Ha, T.H., Muralidharan, S., Bae, J.H., Ha, Y.C., Lee, H.G., Park, K.W. and Kim, D.K.: 2005, Effect of unburnt carbon on the corrosion performance of fly ash cement mortar, Construction and Building Materials, 19, 509–515. https://doi.org/10.1016/j.conbuildmat.2005.01.005

·       Ehsan Ghafari, Seyedali Ghahari, Dimitri Feys, Kamal Khayat, Aasiyah Baig, Raissa Ferron. Admixture compatibility with natural supplementary cementitious materials, Cement and Concrete Composites, Volume 112, 2020, 103683, https://doi.org/10.1016/j.cemconcomp.2020.103683

·       S. Lim, W. Lee, H. Choo, C. Lee, Utilization of high carbon fly ash and copper slag in electrically conductive controlled low strength material, Construction and Building Materials, Volume 157, 2017, Pages 42-50, https://doi.org/10.1016/j.conbuildmat.2017.09.071

Line 84: Nested references [17-21]. Could you please explain what is said in such references?

Line 194: ”3. Results”. Discussion is missing. You should compare and discuss your work with the one performed by other authors. Discussion should be deep. Some references should be mentioned in the discussion, and they should be discussed in deep.

Please, check similar results in the literature and discuss them.

Line 364: Conclusions should highlight the novelty of the work. Please, rewrite the conclusions.

RECOMMENDATION

In conclusion, Major changes have been proposed.

Author Response

 Dear Reviewer, Thank You for Your insightful review of our work, which contributed to a better understanding of the scientific problems related to the subject of the publication and will help with the elimination of potential errors in the future.We would also like to express our gratitude for the revision of our manuscript and the opportunity to re-submit it, incorporating all of the Referees’ suggestions. Our comments and changes are noted below, and are marked in yellow in the manuscript.

Query 1: Lines 98-100: Could you please add the testing methods?

Answer 1: Thank you for your comment. Data have been completed in the text and highlighted in yellow.

Query 2: Line 104: Could you please add the reference (EN 197-1)? EN 197-1:2011. Cement - Part 1: Composition, specifications and conformity criteria for common cement. European Committee for Standardization (CEN): Brussels, Belgium, 2011.

Answer 2: Thank you for your comment. Data have been completed in the text and highlighted in yellow.

Query 3: Line 104:

Data in Table 3. “Chemical composition of cement” should not be a range of values.

Calcium sulfate in Table 3 should be expressed as SO3.

Chemical composition of cement should be expressed as percentage of CaO, SiO2, Fe2O3, Al2O3, and so on.

Could you please add the testing methods?

Answer 3: Thank you for your comment. Data have been completed in the text and highlighted in green.

Query 4: Line 126: thesamples.

Answer 4: Thank you for your comment. Text has been changed and highlighted in yellow.

Query 5: Lines 107-110: This is obvious. It can be deleted.

Answer 5: Thank you for your comment. Text has been removed.

Query 6: Line 124: C content in Fig 4b is about 6%. In particular, a high amount of unburned carbon is responsible for the loss on ignition represents an undesirable constituent of some ashes to be utilized in the reinforced concrete construction. Therefore, it should be mentioned in the introduction. The problem is that the unburned carbon in slags and ashes has several detrimental effects on the concrete. Especially, it increases the electrical conductivity of the concrete, changes the color of mortar and concrete (they may appear black), etc. Moreover, the water/(cement+addition) ratio, needed to obtain a cement paste with a required rheological properties or consistency, is higher forsome slag or ashes with a high carbon content, increasing the corrosivity of metallic parts incorporated in the concrete. Finally, it causes a poor air entrainment behavior and mixture segregation. The following papers deal with this topic:

Freeman, E., Gao, Y-M., Hurt, R. and Suuberg, E.: 1997, Interactions of carbon-containing fly ash with commercial air-entraining admixtures for concrete, Fuel, 76, no. 8, 761–765. https://doi.org/10.1016/S0016-2361(96)00193-7

Ha, T.H., Muralidharan, S., Bae, J.H., Ha, Y.C., Lee, H.G., Park, K.W. and Kim, D.K.: 2005, Effect of unburnt carbon on the corrosion performance of fly ash cement mortar, Construction and Building Materials, 19, 509–515. https://doi.org/10.1016/j.conbuildmat.2005.01.005

Ehsan Ghafari, Seyedali Ghahari, Dimitri Feys, Kamal Khayat, Aasiyah Baig, Raissa Ferron. Admixture compatibility with natural supplementary cementitious materials, Cement and Concrete Composites, Volume 112, 2020, 103683, https://doi.org/10.1016/j.cemconcomp.2020.103683

  1. Lim, W. Lee, H. Choo, C. Lee, Utilization of high carbon fly ash and copper slag in electrically conductive controlled low strength material, Construction and Building Materials, Volume 157, 2017, Pages 42-50, https://doi.org/10.1016/j.conbuildmat.2017.09.071

Answer 6: Thank you for your comment. Discussion has been introduced in the text and highlighted in green.

Query 7: Line 84: Nested references [17-21]. Could you please explain what is said in such references?

Answer 7: Thank you for your comment. Data have been completed in the text and highlighted in green.

Query 8: Line 194: ”3. Results”. Discussion is missing. You should compare and discuss your work with the one performed by other authors. Discussion should be deep. Some references should be mentioned in the discussion, and they should be discussed in deep.

Please, check similar results in the literature and discuss them.

Answer 8: Thank you for your comment. Discussion has been introduced in the text and highlighted in green.

Query 9: Line 364: Conclusions should highlight the novelty of the work. Please, rewrite the conclusions.

Answer 9: Thank you for your comment. Conclusions have been completed in the text and highlighted in green.

Yours sincerely,

Prof of PUT Agnieszka Ślosarczyk

Reviewer 3 Report

The authors replied to all comments and the paper can be accepted 

Author Response

Thank you for aproving our article.

Yours sincerely,

Prof of PUT Agnieszka Ślosarczyk

Reviewer 4 Report

it can be accepted now

Author Response

Thank you for approval our article.

Yours sincerely,

Prof of PUT Agnieszka Ślosarczyk